# Non-Linear Stability of the Step-Variable In-Plane Functionally Graded Plates Subjected to Linear Approaches of the Edges

**DOI:** 10.3390/ma13061439

**Published:** 2020-03-21

**Authors:** Zbigniew Kołakowski, Leszek Czechowski

**Affiliations:** Department of Strength of Materials, Lodz University of Technology, 90-924 Lodz, Poland; zbigniew.kolakowski@p.lodz.pl

**Keywords:** step-variable gradation material, post-critical state, Byskov-Hutchinson’s theory, finite element method

## Abstract

The analysis of gradations through the thickness in structures are commonly used. It usually refers to the problems of the stability of functionally graded (FG) structures. In this work, rectangular in-plane FG plates built of a material gradation along the transversal direction were assumed. Five-strip FG plates with four cases that were based on the boundary conditions on longitudinal edges and simply supported on transverse loaded edges were considered. The non-linear stability problems of the FG plates that were subjected to linear approaches of the transverse edges for several types of loads were solved. The estimations were executed with two methods: an analytical-numerical way based on Koiter’s theory and finite element method (FEM).

## 1. Introduction

Since the 1980s, Functionally Graded Materials (FGMs) have been treated as a new class of composites mainly built of two components. In most cases, there are metallic constituents and ceramic constituents. FGM structures are commonly used in thin-walled construction elements thanks to high thermal properties. A pretty extensive review of theories related FGMs was included in references [1,2,3]. Additionally, in [4], a literature review concerning FG plate structures and analysis of the post-buckling behaviour for a step-variable FG box across the plate thickness is included. At present, different techniques of producing FGMs can be distinguished [5]. 

Owing to that fact, many analytical, numerical, and experimental studies of FGM structures have been conducted to better understand their behaviour with respect to nonlinear stability and limitations of carrying-load capacity in references [6,7,8]. The analysis of vibrations of FGM structures was performed in references [9,10], among others. In works [11,12], constitutive relations referred to neutral physical surface for functionally graded materials based on nonlinear six-parameter shell theory. A response of FGM shell structures under thermal load within the first-order shear deformation theory (FSDT) was investigated in reference [13]. The behaviour of the FGM plate subjected to in-plane compressive or thermal load and combination of both loads, on the basis of the classic plate theory (CPT), was studied in [14]. In reference [15], an analysis of free vibrations of printed polymer FG plate by employing Higher Order Shear Deformation (HSDT) with consideration of changing in the stiffness and density of the material was presented. Analyses of FGM structures applying isogeometric analysis (IGA) were presented in [16,17].

Non-linear stability of step-variable in-plane functionally graded (FG) square plates is presented in [18]. In this work, the stability of plates that were subjected to a shear and compression were considered. Moreover, a post-buckling state of FG plate for uniform approaches of the plate edges were analysed. In a comparison of BHT and finite element method (FEM), pretty close results have been obtained. The present paper is a continuation of [18], in which methods how to solve the problem were discussed in detail. In reference [19], a post-buckling study of step-variable FG cylindrical panels was presented. In reference [20], by applying Koiter’s approach, post-buckling states of orthotropic structures that were made of uniform strips due to compression were attained. 

Referring to accessible literature authors know about works in which a stability of FGM plate structures with a gradation along a plate thickness was considered. The employed method is based on Koiter’s approach with consideration of Byskov–Hutchinson’s theory (BHT) and it has been developed for more than 30 years. The software based on mentioned methods allows for solving the non-linear problem of interaction buckling in the thin-walled structures, which can be composed of isotropic and orthotropic materials as well as of laminates [20,21,22]. The thin-walled plate structures were taken into account. A shear lag phenomenon and distortional deformations were in analyses were taken into account. Simply supports of plates were assumed. It was considered that the assumed material obeyed Hooke’s law. A thin-walled plate was divided into plate elements. For each component of plate, geometrical equations were derived to observe the full bending of the plate. 

In the developed software, the modified method numerically integrating the equations of equilibrium in transverse direction by employing the Runge–Kutta formula with the Godunov’s orthogonalization method has been used [20,21,22]. An accurate method of transition matrix in BHT can be applied in a comparison to the finite strip method (FSM). 

The present work is the further development of the analyses on the basis of FEM to relate the used BHT and to compare the post-buckling curves. It should be noticed that BHT is absolutely faster than FEM. In BHT, the one-mode approach is considered, which is based on adopting a model with one degree of freedom. Thus, a modulation of buckling modes within a load increase was not considered. 

In present paper, the post-buckling equilibrium paths of five rectangular gradation along the transversal direction of plate for five cases of linear approaches of the transverse edges were analysed. Simple supported loaded edges and four manners of support for longitudinal edges were considered. The problem of non-linear stability using two methods was estimated. The first method regarded both a modal solution and a non-linear stability by employing the first- or second-order approximation of Koiter’s approach [18,20,23]. The second way was based on FEM. The present analysis only relates to mechanical load.

## 2. Description of the Problem

The materials of in-plane FGM plate strips obeying Hooke’s law were assumed. For the plates, nonlinear geometrical strain relations were assumed to take the full bending of the plate into account [19,20]:(1)$$\begin{array}{l} \varepsilon_{x}^{}=u_{,x}+{\frac{1}{2}}_{}(w_{,x}^{2}+v_{,x}^{2}+u_{,x}^{2})\\ \varepsilon_{y}^{}=v_{,y}+{\frac{1}{2}}_{}(w_{,y}^{2}+u_{,y}^{2}+v_{,y}^{2})\\ 2\varepsilon_{xy}^{}=\gamma_{xy}^{}=u_{,y}+v_{,x}+w_{,x}w_{,y}+u_{,x}u_{,y}+v_{,x}v_{,y} \end{array}$$
and
(2)$$\kappa_{x}=-w_{,xx}\;\kappa_{y}=-w_{,yy}\;\kappa_{xy}=-2w_{,xy}{}_{}^{}$$
where: *u, v, w*—components of the displacement vector of the panel along the *x, y, z* axis direction, respectively; *ε_x_, ε_y_, ε_xy_* – in-plane strains (full Green’s strain tensor); *κ_x_, κ_y_, κ_xy_*—plate rotations (i.e., the classical laminate plate theory; CLPT). The issue of the nonlinear stability of thin structures was estimated with Koiter’s theory [20,21,22]. Based on all variants of Koiter’s approach [23], the most used way is the method that was developed by Byskov and Hutchinson [20,21,22] formulated in a convenient way. 

Regarding the solution of the problem, BHT in a range of the first and second order approximation was applied. This procedure of estimations is widely described in references [18,20]. For the thin-walled structures with imperfections (defined as a mode of linear buckling), the equations for uncoupled buckling are written in the form [20,21,22]:(3)$$\left(1-\frac{\varepsilon }{\varepsilon_{cr}}\right)\xi +b_{1111}\xi_{}^{3}=\frac{\varepsilon }{\varepsilon_{cr}}\xi_{}^{*}$$
where: ε is the applied strain, $\varepsilon_{cr}$ the buckling strain of the buckling mode, $\zeta_{}^{*}$ the dimensionless amplitude as initial deflection referred to the buckling mode, and finally ζ the dimensionless amplitude of the deflection. The coefficient $b_{1111}$ can be defined based on the equations given in references [20,21,22].

## 3. Analysis of the Calculations Results

The numerical computations for rectangular plates were carried out. The plate that was composed of five strips with the same width along the transverse direction was taken into account (Figure 1a). Table 1 sorts out the material properties for each strip. Different stiffness determined each strip. The assumed geometric dimensions of the plate and a sequence of the strip composition are illustrated in Figure 1. Five types of loading were assumed to be analysed. A linear displacement of transverse edges was assumed (as shown in Figure 1). Type 1 refers to uniform displacements of the loaded edges taken into consideration in [18]. Two next types of loading concern a displacement of edges varying triangularly, however type 2 corresponds to the maximal strain on the ceramic side Al_2_O_3_, whereas type 3 relates to the maximal strain on the aluminium side Al. The strains are equal to 1 in other points of the plate corners (Figure 1b). 

The two last types refer to loading with differ directions of strains, nevertheless the absolute values of the strains on the opposite edges are the same. In the case of type 4, the strip Al_2_O_3_ is stretched, but strip Al is compressed. For type 5, it is reverse (see Figure 1).

In addition, four cases of boundary conditions were assumed: simply supported plate on all of its edges and a mixture of simply supported and clamped edges. With regard to the plates that are built of strips with different stiffness, the following notation for a sequence of the boundary conditions was introduced: line (1) for x = 0, line (2) for y = 0, line (3) for x = L, and line (4) for y = b. The following boundary conditions were marked: SSSS, SCSC, SSSC, and SCSS, where S—simply supported edge, C—clamped edge. 

Four cases of the boundary conditions were taken into account for the study of non-linear stability, namely:Case A – SSSS: $\begin{array}{ll} w(x=0)=w(x=L)=0; & M_{x}(x=0)=M_{x}(x=L)=0;\\ u(x=0)=u(x=L)=const; & v(x=0)=v(x=L)=0;\\ w(y=0)=w(y=b)=0; & M_{y}(y=0)=M_{y}(y=b)=0;\\ v(y=0)=v(y=b)=0; & N_{xy}(y=0)=N_{xy}(y=b)=0; \end{array}$Case B – SCSC:$\begin{array}{ll} w(x=0)=w(x=L)=0; & M_{x}(x=0)=M_{x}(x=L)=0;\\ u(x=0)=u(x=L)=const; & v(x=0)=v(x=L)=0;\\ w(y=0)=w(y=b)=0; & w_{,y}(y=0)=w_{,y}(y=b)=0;\\ v(y=0)=v(y=b)=0; & N_{xy}(y=0)=N_{xy}(y=b)=0; \end{array}$Case C – SSSC: $\begin{array}{llll} w(x=0)=w(x=L)=0; & & M_{x}(x=0)=M_{x}(x=L)=0; & \\ u(x=0)=u(x=L)=const; & & v(x=0)=v(x=L)=0; & \\ w(y=0)=0; & M_{y}(y=0)=0; & v(y=0)=0; & N_{xy}(y=0)=0;\\ w(y=b)=0; & w_{,y}(y=b)=0; & v(y=b)=0; & N_{xy}(y=b)=0; \end{array}$Case D – SCSS: $\begin{array}{llll} w(x=0)=w(x=L)=0; & & M_{x}(x=0)=M_{x}(x=L)=0; & \\ u(x=0)=u(x=L)=const; & & v(x=0)=v(x=L)=0; & \\ w(y=0)=0; & w_{,y}(y=0)=0; & v(y=0)=0; & N_{xy}(y=0)=0;\\ w(y=b)=0; & M_{y}(y=b)=0; & v(y=b)=0; & N_{xy}(y=b)=0; \end{array}$

The stress and moment resultants (*N_x_, N_y_, N_xy_, M_x_, M_y_, M_xy_*) are defined while using CLPT, as in references [20,21], among others. 

The buckling strains were determined to compare the post-buckling equilibrium paths for each case. For five cases of load (Figure 1b), a five-strip rectangular FG plate of the dimensions (Figure 1a): $b=200^{}mm$, plate thickness $t=2^{}mm$, and each strip b1=b2=b3=b4=b5=0.2b were assumed. The length of the plate L was different for the considered boundary conditions and determined based on minimal value of bifurcation.

The present results were obtained by two methods:BHT by using the first- and second-order non-linear approximation—more details can be found in references [18,20]; and,FEM with the commercial ANSYS^®^ software [24]—a discrete model is illustrated in Figure 2.

The element was considered to model a strip plate [24]. The element is characterised by eight nodes with six degrees of freedom in each node. The used element is pretty good for linear and non-linear simulation for both isotropic and multilayers structures. This element is based on Mindlin–Reissner’s theory (the first-order deformation theory). Figure 2 shows a discrete model with the conditions of simulations. The size of the finite element amounted to 2 mm long. The numerical calculations were carried out for large displacements with Green–Lagrangian equations. The computations were performed in steps by setting a number of steps from 100 to 500. Beam189 elements with high stiffness on the loaded edges were taken into consideration on the loaded edges to satisfy type 4 and type 5 of loads. The assumption of the beam elements on these edges does not influence the total bending stiffness of the plate, because all of edges of the plate were constrained in the perpendicular direction with respect to the in-plane of the plate. The parameter Δ that is given in Figure 2 denotes a displacement of the point or the edge.

### 3.1. Analytical-Numerical Method Using BHT

This subsection presents the results that were obtained within BHT. Table 2 lists the values of bifurcational strains for the considered cases and types of load. For the cross-section area of the plate for x=const (Figure 1), a displacement of the gravity centre from axis $y=b/2=100^{}mm$ due to different density of individual strips (Table 1) occurs. The coordinate of the gravity centre in the cross-section area of the plate is $y_{c}=95.41mm$. The bifurcational load acting in the plate was reduced to this central point. In Table 2, the values of the buckling forces $P_{cr}$ and the buckling bending moments $M_{cr}$ acting in the plate plane were given. A positive value of the moment is assumed when the ceramic strip is compressed, whereas the metal strip stretched. 

A negative value $P_{cr}$ denotes the tension force, however a negative value of the moment $M_{cr}$ means an opposite value of the moment. An influence of the boundary conditions on the stability and post-buckling equilibrium paths was compared for each of five types of load. As is known from the literature, the lowest values of bifurcational loads are for **A** (SSSS), however they are the greatest ones for **B** (SCSC). For the case **C** (SSSC) and **D** (SCSS), the critical loads are intermediate. 

The strip that is built of the FG plate can introduce some kind of modification, because the ceramic strip is significantly stiffer than the aluminium one. For type 1, the above formulated rules for the buckling strains are concerned. The longest plate L is for the case **A** but the shortest one is for case **B**. In the case **C**, the bifurcational values are lower than for the case **D**, but it is opposite in the case of the plate length L. Conclusions are very similar for types 2 and 3. The exception is type 3 for the case **C** and **D**. Here, a contrary sequence of bifurcational values occurs. In type 3 for the case **D**, the stiffest ceramic strip is less loaded and the condition of its fixing does not matter meaningfully as the fixing of the aluminium strip at minor stiffness. Thus, a change in bifurcational values follows due to this reason. The critical values of moments are low and negative, which points out to a change in direction. For type 4, the ceramic strip is elongated, but the aluminium strip is compressed, which decides about the stability loss of the plate for the assumed boundary conditions. The bifurcational values for the cases **A** and **D** are the lowest and close to each other, similarly as the length L; however, for cases **B** and **C**, the values are the highest and the same refers to the length. The assumed distribution of strains for type 4 explains the negative values of forces and moments that are caused by considerable changes in the stiffness of utmost strips. The negative force means that the plate is stretched, but a negative moment results from a change of its direction of operation. For type 5, the ceramic strip is compressed, which decides about the loss stability, but the aluminium strip is stretched. For cases **A** and **C**, the lowest critical values are achieved contrary to the values of the cases **B** and **D**, which are the highest and differ slightly from each other. The lengths L for the cases **A** and **C** are greater than for the cases **B** and **D**. In Figure 3, Figure 4, Figure 5, Figure 6 and Figure 7, post-buckling paths (Equation (3)) for the FG plates subjected to five types of loads 1–5 are shown. On each of the charts, post-buckling curves for the cases **A–D** are depicted. 

The post-buckling paths for the perfect plates (i.e., for $\zeta^{*}=0.0$ in Equation (3)) are presented in an untraditional system of coordinates, namely $\varepsilon /\varepsilon_{cr}$ as a function of the non-dimensional square amplitude of the deflection $\zeta^{2}$, which causes the post-buckling paths to be straight lines, according to Equation (3). For types 2 (Figure 4), 3 (Figure 5), and 5 (Figure 7), the lowest post-buckling equilibrium paths respond to the case **C** (SSSC), next **D** (SCSS) and the path for **B** (SCSC) and the highest path for **A** (SSSS). For type 1 (Figure 3), the lowest one is for **C**, next for the tapes **A** and **D**, and the highest one for **B**. In Figure 6, for type 4, the order of the paths is as follows: **D**, **C**, **B**, and **A**. The type of load and the boundary conditions on the stiffest edge determine the order of the post-buckling paths (i.e., strip 1).

### 3.2. Comparison of the BHT Results with FEM Results

On the basis of the conclusions resulting from [18], which confirms close correlation of the results from the BHT and the FEM for a uniform displacement of the plate edges, the BHT results were verified by FEM in 25 percent, as shown in the last column of Table 2 in the present work. A special attention was paid to types 4 and 5. As it can be easily seen, very close results were received with both the methods. Figure 8, Figure 9, Figure 10, Figure 11 and Figure 12 present the post-buckling equilibrium paths for the verified cases in a traditional system of coordinates $\varepsilon /\varepsilon_{cr}$ as a function of Δw/t for the imperfection $\zeta^{*}=w_{0}/t=0.1$. In general, the paths go through similarly, though some greater discrepancies are visible for the boundary condition A, type 4 (Figure 9). For the remaining variants, differences in the plate deflections turn out in the vicinity of the buckling load (Figure 8) or at circa $1.5\cdot \varepsilon /\varepsilon_{cr}$ (Figure 10, Figure 11 and Figure 12). It should be noted that the BHT results were obtained for the uncoupled buckling mode (the modulation of buckling modes within a load increase was not considered). It can be reason that some small differences between FEM and BHT appear.

## 4. Summary

The analysis of non-linear stability of graded plate that was subjected to linear approaches of the transverse loaded edges was carried out. Four cases of the supports and five different loads were assumed. The estimations were conducted mainly based on Koiter’s approach with the use of BHT. The second used method was relied on FEM by applying commercial ANSYS^®^ software. The obtained results with these two methods showed good correlation.

## Figures and Tables

**Figure 1 materials-13-01439-f001:**
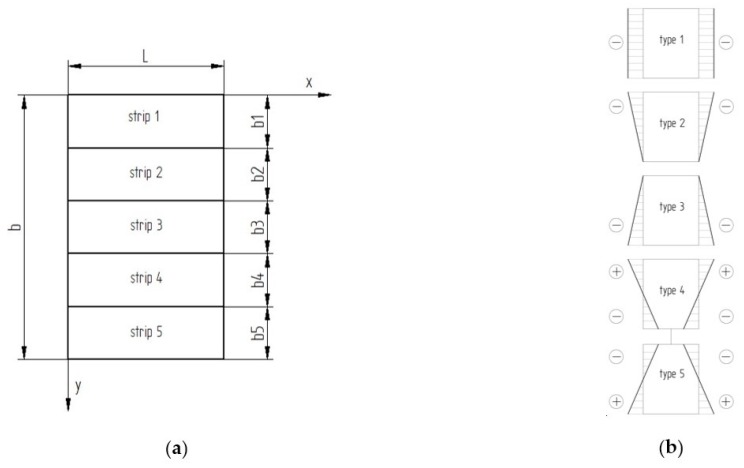
In-plane functionally graded (FG) rectangular plate with coordinate system (**a**) and types of assumed loads (**b**).

**Figure 2 materials-13-01439-f002:**
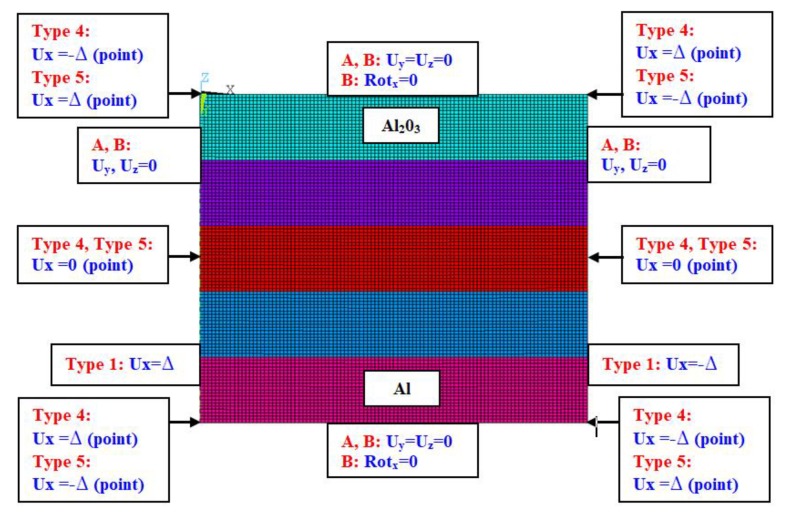
Numerical model of the rectangular plate with boundary conditions.

**Figure 3 materials-13-01439-f003:**
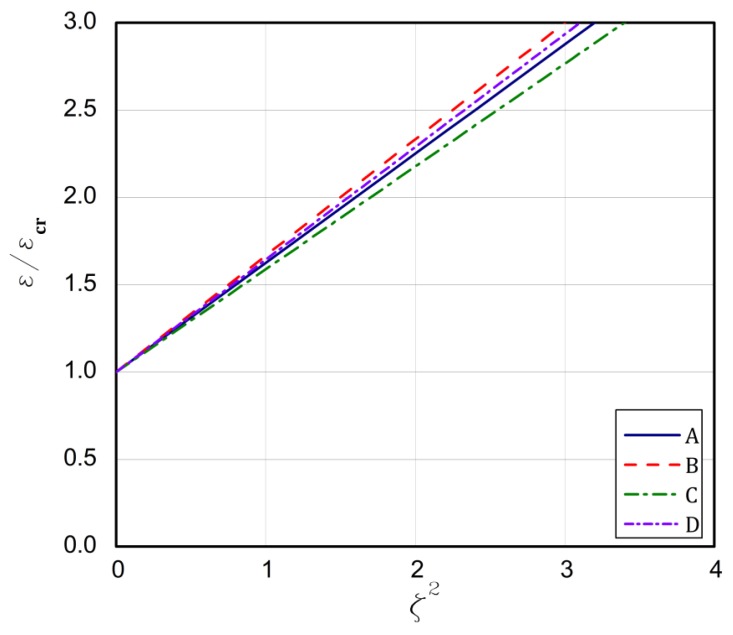
The post-buckling paths $\varepsilon /\varepsilon_{cr}$ in a function of $\zeta^{2}$ for type 1 of loads of the perfect plates.

**Figure 4 materials-13-01439-f004:**
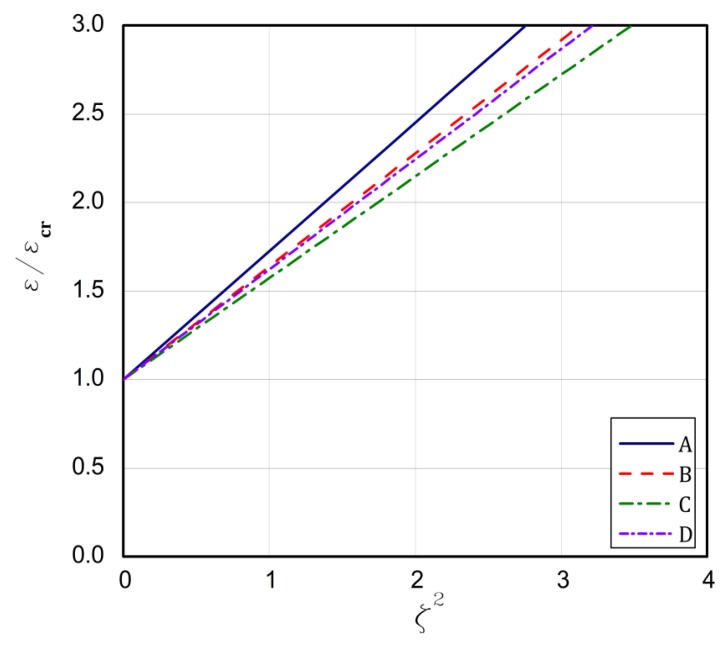
The post-buckling paths $\varepsilon /\varepsilon_{cr}$ in a function of $\zeta^{2}$ for type 2 of loads of the perfect plates.

**Figure 5 materials-13-01439-f005:**
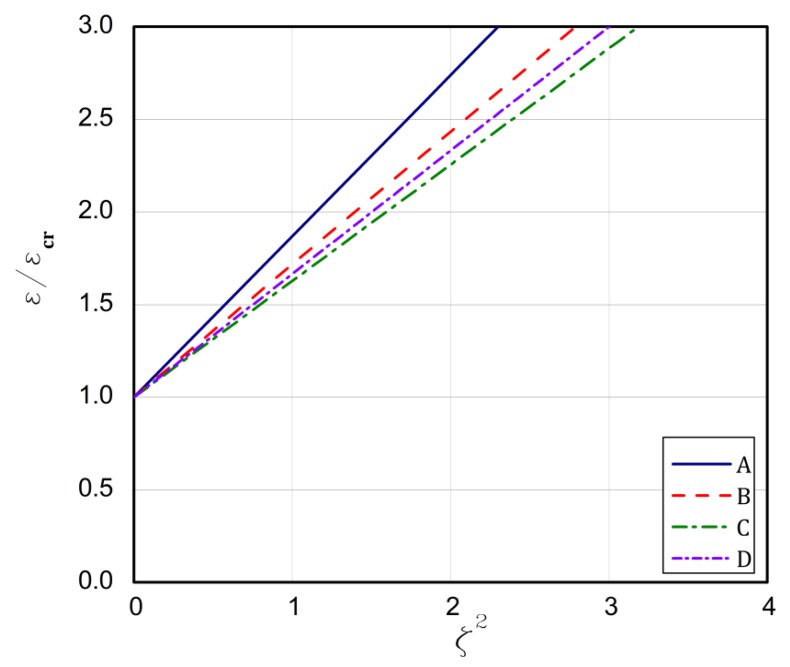
The post-buckling paths $\varepsilon /\varepsilon_{cr}$ in a function of $\zeta^{2}$ for type 3 of loads of the perfect plates.

**Figure 6 materials-13-01439-f006:**
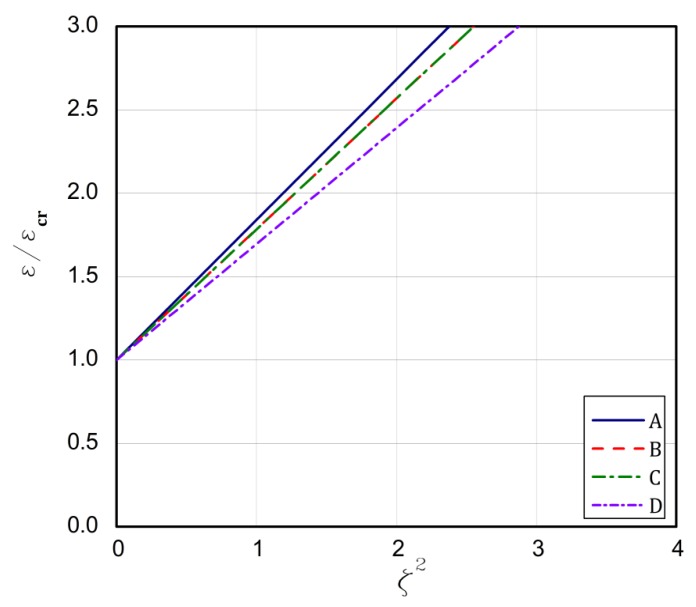
The post-buckling paths $\varepsilon /\varepsilon_{cr}$ in a function of $\zeta^{2}$ for type 4 of loads of the perfect plates.

**Figure 7 materials-13-01439-f007:**
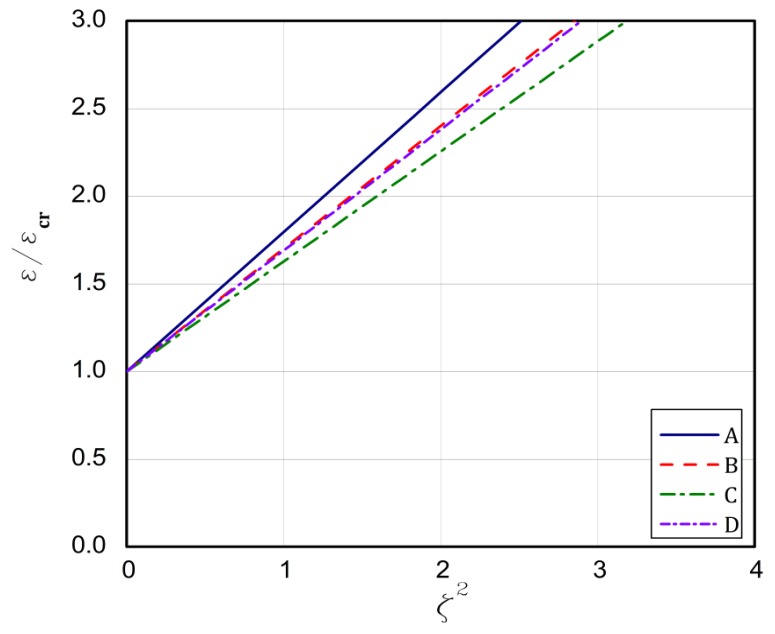
The post-buckling paths $\varepsilon /\varepsilon_{cr}$ in a function of $\zeta^{2}$ for type 5 of loads of the perfect plates.

**Figure 8 materials-13-01439-f008:**
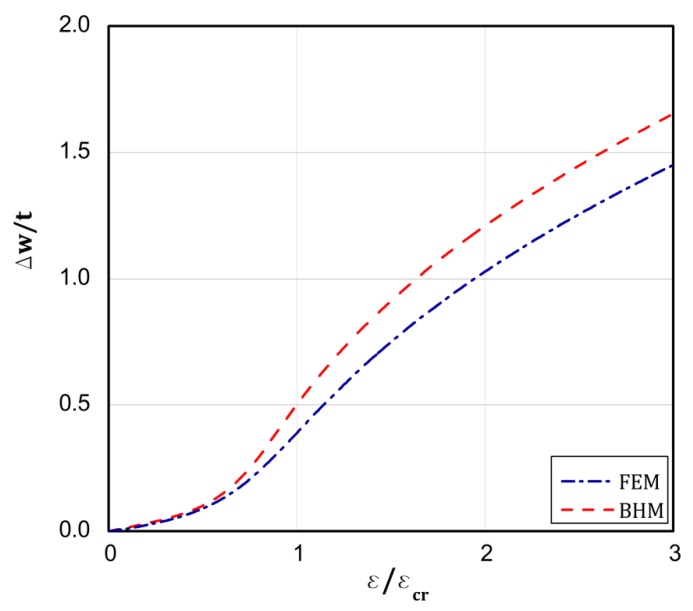
Dependence of Δw/t vs. $\varepsilon /\varepsilon_{cr}$ for variant **A** and type 1 of loads.

**Figure 9 materials-13-01439-f009:**
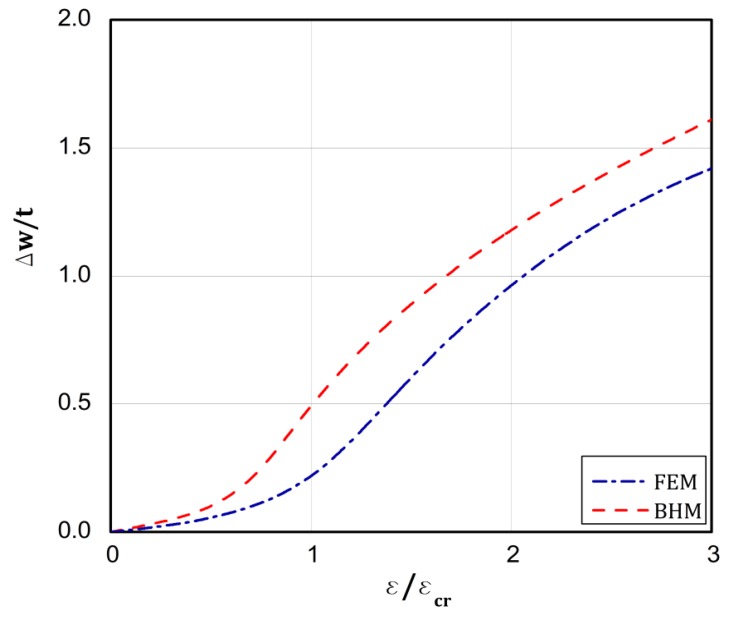
Dependence of Δw/t vs. $\varepsilon /\varepsilon_{cr}$ for variant **A** and type 4 of loads.

**Figure 10 materials-13-01439-f010:**
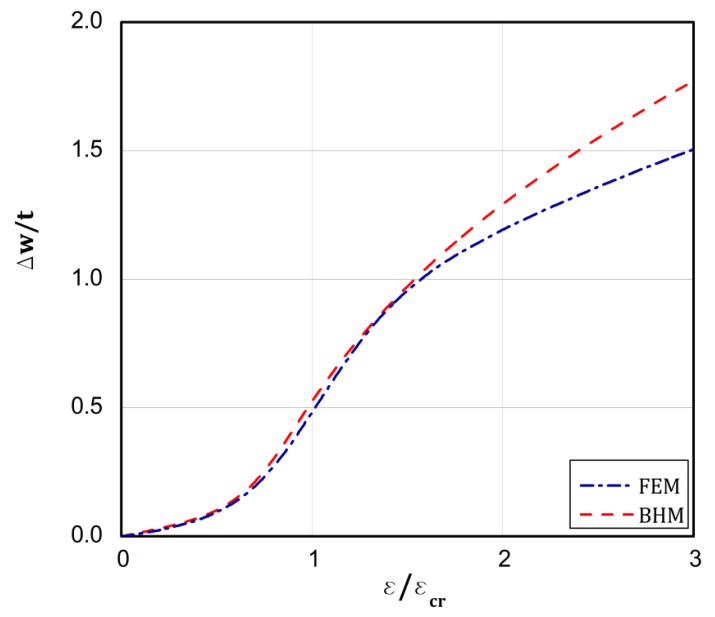
Dependence of Δw/t vs. $\varepsilon /\varepsilon_{cr}$ for variant **A** and type 5 of loads.

**Figure 11 materials-13-01439-f011:**
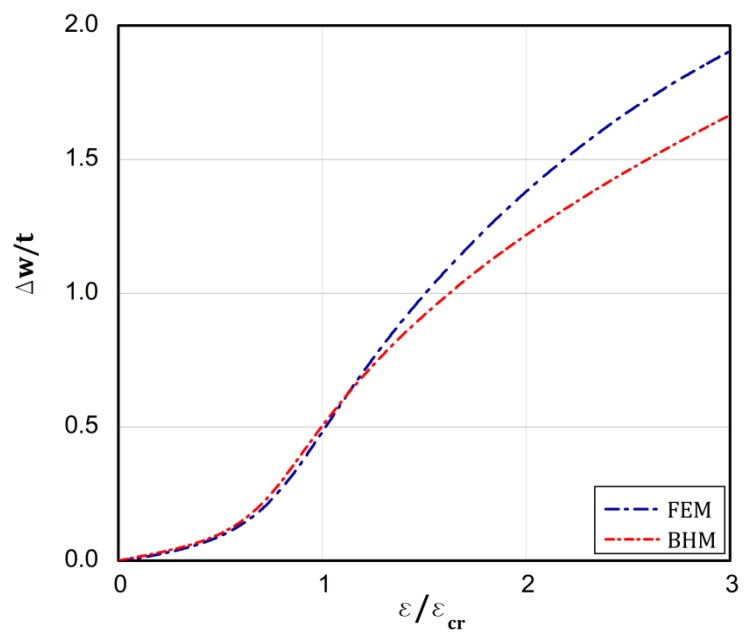
Dependence of Δw/t vs. $\varepsilon /\varepsilon_{cr}$ for variant **B** and type 4 of loads.

**Figure 12 materials-13-01439-f012:**
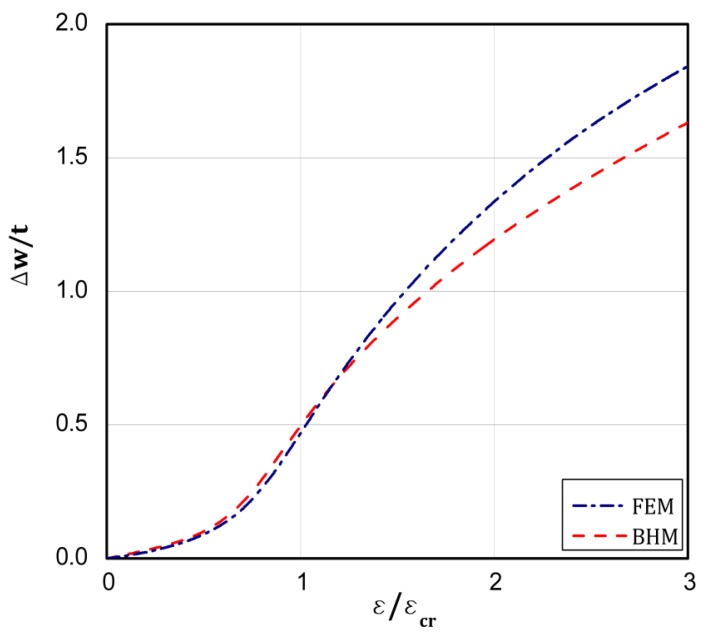
Dependence of Δw/t vs. $\varepsilon /\varepsilon_{cr}$ for variant **B** and type 5 of loads.

**Table 1 materials-13-01439-t001:** Mechanical properties for detailed strips.

Number of Strip	Description	Young’s Modulus E (GPa)	Poisson Ratio Ν (-)
1	Al_2_O_3_ (100%)	393	0.25
2	Al_2_O_3_ (75%) + Al (25%)	312	0.27
3	Al_2_O_3_ (50%) + Al (50%)	231	0.29
4	Al_2_O_3_ (25%) + Al (75%)	151	0.31
5	Al (100%)	70	0.33

**Table 2 materials-13-01439-t002:** Results of the lowest buckling values.

Boundary Conditions	Type of Load	$\varepsilon_{cr}\mathrm{BHT}$	L mm	$P_{cr}\mathrm{kN}$	$M_{cr}\mathrm{kNm}$	$\varepsilon_{cr}\mathrm{FEM}$
**A**—SSSS	1	0.34089 × 10^−3^	204	31.55	0.7354	0.349 × 10^−3^
	2	0.63723 × 10^−3^	204	37.71	1.633	-
	3	0.70014 × 10^−3^	198	23.36	−0.2837	-
	4	0.23685 × 10^−2^	125	−61.16	−7.029	0.235 × 10^−2^
	5	0.19076 × 10^−2^	150	49.26	5.661	0.195 × 10^−2^
**B**—SCSC	1	0.59344 × 10^−3^	136	54.92	1.280	-
	2	0.11492 × 10^−2^	134	68.02	2.945	-
	3	0.11618 × 10^−2^	133	38.76	−0.4708	-
	4	0.35509 × 10^−2^	96	−91.69	−10.53	0.357 × 10^−2^
	5	0.34002 × 10^−2^	96	87.80	10.09	0.338 × 10^−2^
**C**—SSSC	1	0.43339 × 10^−3^	172	40.11	0.9350	-
	2	0.75476 × 10^−3^	174	44.67	1.934	-
	3	0.96399 × 10^−3^	165	32.16	−0.3906	-
	4	0.35509 × 10^−2^	96	−91.69	−10.53	-
	5	0.19166 × 10^−2^	146	49.49	5.688	-
**D**—SCSS	1	0.48812 × 10^−3^	154	45.18	1.053	-
	2	0.10251 × 10^−2^	150	60.67	2.627	-
	3	0.88588 × 10^−3^	154	29.56	−0.3590	-
	4	0.23713 × 10^−2^	125	−61.23	−7.038	-
	5	0.33999 × 10^−2^	97	87.79	10.09	-

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
