# Peer review of "Non-Linear Stability of the Step-Variable In-Plane Functionally Graded Plates Subjected to Linear Approaches of the Edges"

_materials, 2020, doi:10.3390/ma13061439_

Round 1

Reviewer 1 Report

Line 13: it is length-wise problem
in-plane means the variation of materials in both x and y directions.

Introduction must be improved. please refer to other researcher 's work and explain their findings and discuss why your approach is the most suitable one:

good ref for FGM (vibration)

M Amirpour, S Bickerton, E Calius, BR Mace, R Das, 2019, Numerical and experimental study on free vibration of 3D-printed polymeric functionally graded plates, Composite Structures 189, 192-205

Line 50: which software, please explain it.

Line 60: Explain in detail and reference for the used method.

Line 81: why the mentioned theory is the most used?

Line 149: the theory should explain in introduction with ref.

Line 151: in numerical, how author get their result converged? did they study convergence?

All Figures 3, 4 and 5:  please explain the graphs in detail as they are not clear and what it is showing?

In table 2: whay there is no data for e_cr for some cases?

Line 222: 25% error? what are the contributing factors leading the error?

Author Response

Dear Reviewer,

first of all, we’d like to thank for your valuable remarks, which let us improve our paper. Taking into consideration these suggestions, we have done our best to refine our article and to fulfil the requirements of publishable standard. The added/changed text was highlighted on “yellow”. In case of acceptance before publishing, the English in manuscript will be checked once again by Proof reader.

With best Regards,

Zbigniew Kolakowski

Leszek Czechowski

Reviewer 2 Report

Comment for materials-747202 is listed as follows,

  1. There are some miss been named, referred and error typing.

(1) In the Abstract, please change the "Application of gradations…" into " Analysis of gradations…", because the "application (hardware)" is not found in the manuscript.

(2) In the Keywords, please check the "functionally step-variable graded material", "post-buckling state" and "asymptotic Koiter’s theory",  they are missed in the manuscript.

(3) In page 5, line 164, please change the " Byskov and Hutchinson’s theory (BHT)" into " BHT ", because it is already defined in the manuscript.

(4) In the section "2. Formulation of the problem", please change it into "2. Description of the problem", because there are not formulation delivered.

(5) In the eqs. (1) and (2), please define all the variable names, e.g. " ,  and are in-plane strains." etc.

(6) In line 78, the equation type of displacements should be mentioned, e.g. the first-order shear deformation theory (FSDT) or third-order shear deformation theory (TSDT). Also "u, v, w" are not in the same form of letters with that parameters in the eqs.

(7) In the eq. (3) and Figure 1(a), please check and define the variable "b", because it is defined in duplicate with the same name. Also define for ''L" in Figure 1(a), e.g. "L is the length of plates in the x direction." in the manuscript.

(8) In pages 3-4, lines 115- 133, please define the variable names " ", " ", " ", " " etc. they did not been defined. Check " ", it is not correct, eg. " " is collect.

(9) In line 136, please check the , because  did not been defined.

(10) In lines 49 and 140, please check for the abbreviation name "BHT", because the "Byskov-Hutchinson’s theory (BHT)" and "‘full’ Byskov and Hutchinson’s theory (BHT)" are not in the same name.

(11) In line 163, please change the "3.1. Analytical-numerical method using Byskov-Hutchinson’s theory (BHT)" into the " 3.1. Analytical-numerical method by using BHT".

(12) In line 164, please change the " Byskov and Hutchinson’s theory (BHT)" into the " BHT", because it had been defined in line 49.

(13) In line 139, please change the " The results were obtained by two methods: " into " The present results were obtained by two methods: ", because they are under your presentation work.

(14) In line 173, in title name of Figure 3, please change the "…for type 1 of the perfect plates." into " …for type 1 of loads of the perfect plates.". Same change also for Figures 4-12.

(15) Next line 207, in column name of Table 2, please change the " Example of load" into " Type of load".

(16) For the "four cases" of boundary conditions, in line 110, please change the " A-SSSS, B-SCSC, C-SSSC, D-SCSS" into " SSSS, SCSC, SSSC, SCSS"; in line 114, please change the " A – SSSS" into " Case A – SSSS"; in line 119, please change the " B – SCSC" into " Case B – SCSC"; in line 124, please change the " C – SSSC" into " Case C – SSSC"; in line 129, please change the " D – SCSS" into " Case D – SCSS".

(17) In line 142, please change " with the ANSYS® software [22] based on the finite element method–" into " with the commercial ANSYS® software [22] based on the FEM–". Please delete the lines 144-145 sentences " Based on a good agreement of the results provided by both methods as in reference [16], the FEM calculations were limited to a few cases."

(18) In lines 251-252, please change " The problem was solved within the first- and second-order approximation of Koiter’s theory and FEM." into " The problem was solved within the first- and second-order approximation of Koiter’s theory by BHT and FEM by commercial ANSYS® software."

Author Response

(The authors gave the same response as above.)

Reviewer 3 Report

The manuscript discusses the modeling of nonlinear stability of functionally graded plates based on a step variable approach. The manuscript is well organized, and the topic falls within the scope of the journal. Before recommending publication, I would like to ask the authors to address the remarks listed below.

(1) In the last sentence before section 3, the authors mentioned that “the coefficient b can be obtained as in the literature”. In order to make the formulation relatively complete, the authors should at least provide a brief explanation of how to obtain the coefficient, instead of simply providing the reference without saying anything.

(2) In the first paragraph under section 3, the authors simplified the continuous grading of functionally graded materials in the thickness direction as five isotropic thickness layers. The authors should perform a sensitivity analysis to see if the assumption of five isotropic thickness layers is sufficient to obtain accurate solutions. Comparison to more layers such as 10 should be made. Otherwise, it is not convincing that the solution is converged.

(3) The mesh of the finite element model as shown in Fig. 2 seems too fine. Please explain why such a large mesh density is used. Moreover, have the authors done convergence study?

(4) In figs 8 and 9, the predictions from the two methods are significantly different, which is not acceptable. Please provide a reasonable explanation. The authors are also suggested to use more thickness layers in the FEM model to see if solution gets better.

(5) A few important publications on the modeling of functionally graded plates and shells as well as stability analysis based on a layered approach have been left out. These include: (a) doi.org/10.1016/j.compstruct.2017.05.037 (b) doi.org/10.1016/j.compstruct.2020.111893

(6) There exists a few typos/grammar errors in the manuscript, such as “the of BHT estimation” and “one mode approach was applied what means an assumption”. Please go through the manuscript and make sure they are fixed.

Author Response

Dear Reviewer,

first of all, we’d like to thank for your valuable remarks, which let us improve our paper. Taking into consideration these suggestions, we have done our best to refine our article and to fulfil the requirements of publishable standard. The added/changed text was highlighted on “yellow”. In case of acceptance before publishing, the English in manuscript will be checked once again by Proofreader.

With best Regards,

Zbigniew Kolakowski

Leszek Czechowski

Round 2

Reviewer 1 Report

thanks for all modifications.

the author successfully amended all required comments.

Reviewer 2 Report

Accept.